# Formulation and Optimization of Metronidazole and *Lactobacillus* spp. Layered Suppositories via a Three-Variable, Five-Level Central Composite Design for the Management of Bacterial Vaginosis

**DOI:** 10.3390/pharmaceutics14112337

**Published:** 2022-10-29

**Authors:** Margaret O. Ilomuanya, Busayo B. Salako, Modupe O. Ologunagba, Omonike O. Shonekan, Kruga Owodeha-Ashaka, Eseosa S. Osahon, Andrew N. Amenaghawon

**Affiliations:** 1Department of Pharmaceutics and Pharmaceutical Technology, Faculty of Pharmacy, University of Lagos, PMB 12003, Surulere, Lagos 101245, Nigeria; 2Department of Pharmaceutical Chemistry, Faculty of Pharmacy, University of Lagos, PMB 12003, Surulere, Lagos 101245, Nigeria; 3Waterford Institute of Technology, School of Science and Computing, X91 K0EK Waterford, Ireland; 4Department of Chemical Engineering, Faculty of Engineering, University of Benin, Benin City 300283, Nigeria

**Keywords:** central composite design, bacterial vaginosis, layered suppositories, *Lactobacillus* spp.

## Abstract

Bacterial vaginosis, a polymicrobial clinical syndrome characterized by a shift in healthy vaginal microbiota due to bacterial colonization, is characterized by high recurrence rates after conventional treatment with an antimicrobial agent. This has necessitated the need to develop a formulation that has the potential to ensure *Lactobacilli* viability and bacterial clearance. This study seeks to develop and optimize a layered suppository using a five-level central composite design to ensure optimized metronidazole release and lactic acid viability. Layered suppositories were formulated using the fusion method using polyethylene glycol blend 1500/4000 and Ovucire^®^ as suppository bases. *Lactobacillus fermentum* was incorporated in the molten mass before molding the solid body suppositories into the cavity of hollow-type suppositories and sealing the molten excipients. Artificial neural network model predictions for product optimization showed high predictive capacity, closely resembling experimental observations. The highest disintegration time recorded was 12.76 ± 0.37 min, with the optimized formulations showing lower times of 5.93 ± 0.98 min and an average weight of 1.17 ± 0.07 g. Histopathological observations determined high compatibility of suppositories with vaginal cells with no distortion or wearing of the vagina epithelium. This optimized formulation provides a safe and promising alternative to conventional suppositories in the treatment and prevention of the recurrence of bacterial vaginosis.

## 1. Introduction

Bacterial vaginosis (BV) remains the commonest vaginal infection, with an approximately 50% recurrence rate among infected women within one year of infection [1]. Some research suggests that it may precipitate preterm labor and has been associated with the development of pelvic inflammatory disease [1]. In Nigeria, a prevalence rate of 17%, 17.3%, and 25% have been reported from separate studies conducted in the south-east, north-east, and south-west Nigeria, respectively [2,3].

According to the Centre for Disease Control and Prevention Sexually Transmitted Disease guidelines, the first line recommended treatment for BV is oral metronidazole, 500 mg twice daily for 7 days [4]. However, metronidazole oral therapy is associated with side effects, including gastrointestinal effects (metallic taste in the mouth, nausea, vomiting) and Candida infection. Since vaginal regimens have been associated with fewer gastrointestinal complaints (33% vs. 52%), vaginal metronidazole may be an alternative to oral metronidazole [5,6]. 

Metronidazole is a nitro imidazole antimicrobial agent used to manage protozoal infections such as trichomonas and anaerobic infections. Metronidazole has been used widely in the treatment of BV with good clinical results [5]. Various preparations allowing vaginal or oral administration and different regimens have been studied [6].

Probiotics indirectly contribute to the treatment of BV, preventing the infections’ recurrence and contagion. Many studies have confirmed that probiotics are effective in the treatment of vaginal infections such as BV by positively altering the intravaginal microbiota composition [7].

*Lactobacillus* spp., a probiotic is defined by the Food and Agriculture Organization and World Health Organization as live microorganisms that, when consumed/administered in adequate amounts, confer beneficial health effects to humans or animals [8]. *Lactobacillus fermentum*, *Lactobacillus casai*, L. acidophilus, and *Lactobacillus iners* are some of the species which have been found in the vagina. *Lactobacillus* spp. ferment glycogen secreted by vaginal epithelial cells into lactic acid, and colonization by these microorganisms correlates to the low pH in the vagina [9].

There is a dearth of studies involving the optimization of suppositories containing both active pharmaceutical ingredients and probiotics via the prediction of their release profile and bioavailability. Traditional experiments require more effort, time, and materials when a complex formulation needs to be developed. Recently, response surface methodology (RSM) via central composite design (CCD) coupled with statistically designed experiments has been found to be very useful in optimizing multivariable processes, and it has been successfully applied to the optimization of many bioprocesses [10]. Hence, the development and optimization of a vaginal suppository containing an antimicrobial agent and a probiotic using responding surface methodology via central composite design would facilitate the prediction of the kinetic release profile and bioavailability of the suppository with a minimum number of trials. Moreover, this formulation would provide a synergistic effect and prevent recurrence in the management of bacterial vaginosis. 

## 2. Materials and Methods

Lactic acid bacillus *(L. acidophilus)* (human vaginal epithelial cells) were obtained from the Nigeria Institute of Medical Research NMIR (Lagos, Nigeria). Polyethylene glycol (PEG) 1500 was obtained from Viswaat Chemicals Ltd. (Mumbai, India), and PEG 4000 was obtained from Medex Pharmaceuticals (Leicester, UK). Ovucire was generously donated by Gattefossé France. Tween 80 was obtained from Gracefruit Ltd. (Longcroft, UK); Sodium citrate was obtained from Runchuang Food Additives Company (Taizhou, China); Metronidazole was obtained from Xi’an Harmonious Natural Bio-tech Ltd. (Xi’an, China); Light Mineral Oil was obtained from Asian Oil Company (Mumbai, India). All other chemicals used were of high analytical grade.

### 2.1. Design of Experiment

A three-variable, five-level central composite design (CCD) was used to plan the experiments using Design Expert^®^ software (version 7.0.0, Stat-ease, Inc. Minneapolis, MI, USA). The CCD was chosen because it is suitable for quadratic response surfaces [11]. In particular, the star points of the CCD are useful in estimating the curvature of nonlinear response surfaces. The independent variables were Ovucire/PEG ratio and surfactant concentration, and they were coded using Equation (1) with values shown in Table 1. The responses measured were drug metronidazole release and *Lactobacilli* viability. The experimental runs were performed in a random manner to minimize the effects of unexplained variability in the response [12,13].
(1)$$x_{i}=\frac{X_{i}-X_{o}}{\Delta X_{i}}$$

In Equation (1), *x_i_* and *X_i_* are the coded and actual values of the independent variables, respectively, while *X_o_* is the actual value of the independent variable at the center point, and Δ*X_i_* is the step change in *X_i_*. 

### 2.2. Artificial Neural Network Modelling

A commercial artificial neural networks (ANN) software, Neural Power (version 2.5, C.P.C-X Software USA), was used to model and optimize the dependent variables (drug metronidazole release and *Lactobacilli* viability) as a function of the independent variables (Ovucire/PEG ratio and surfactant concentration). ANNs are modeled after the human neural system and can build relationships between variables. The neural network modeling procedure followed is like that reported in our earlier works [14,15]. In a sequential manner, the most suitable transfer function, best network architecture, best training algorithm, and optimum number of neurons were determined iteratively. The transfer functions assessed were sigmoid, hyperbolic tangent, gaussian, linear, threshold, linear, and bipolar linear. Two network architectures, namely multilayer normal feed forward (MNFF) and multilayer full feed forward (MFFF), were assessed to determine which was more suitable for modeling the responses. Both network types had two hidden layers for enhanced predictive capabilities. The best training algorithm was selected from among incremental back propagation (IBP), batch back propagation (BBP), quick propagation (QP), generic algorithm (GA), and Levenberg–Marquadt algorithm (LM). In training the networks, the experimental data set was divided into three parts. In total, 70% was used for the actual training, while 15% was used for validation and testing. After establishing the best network architecture with its associated training algorithm and transfer function, the optimum number of neurons in both hidden layers was then determined following an iterative process. The number of neurons is important in the sense that too few neurons lead to poor fitting, while too many neurons result in over fitting [16].

### 2.3. Goodness of Fit Assessment of the ANN Model

The goodness of fit statistical indicators such as correlation coefficient (*R*), coefficient of determination (*R*^2^), adjusted *R*^2^, mean square error (*MSE*), root mean square error (*RMSE*), standard error of prediction (*SEP*), mean absolute error (*MAE*), and average absolute deviation (*AAD*) were used to assess the fit of the ANN model to the experimental observations. Equations (2)–(9) show the expressions for these statistical indicators. The R, R^2^, and adjusted R^2^ values need to be close to one as much as possible. An R^2^ value of one is the ideal case of a perfect fit between experimental and model predicted results. Specifically, the R^2^ value should have a minimum value of 0.8 for there to be an acceptable fit between predicted and experimental results. Conversely, the error terms (*MSE*, *RMSE*, *SEP*, *MAE*, and *AAD*) should be as small as possible [14,16].
(2)$$R=\frac{\sum_{i=1}^{n}\left(x_{p,i}-x_{p,ave}\right)\cdot \left(x_{a,i}-x_{a,ave}\right)}{\sqrt{\left[\sum_{i=1}^{n}{\left(x_{p,i}-x_{p,ave}\right)}^{2}\right]\left[\sum_{i=1}^{n}{\left(x_{a,i}-x_{a,ave}\right)}^{2}\right]}}$$
(3)$$R^{2}=1-\frac{\sum_{i=1}^{n}{\left(x_{a,i}-x_{p,i}\right)}^{2}}{\sum_{i=1}^{n}{\left(x_{p,i}-x_{a,ave}\right)}^{2}}$$
(4)$$Adjusted\;R^{2}=1-\left[\left(1-R^{2}\right)\times \frac{n-1}{n-k-1}\right]$$
(5)$$MSE=\frac{1}{n}\sum_{i=1}^{n}{\left(x_{p,i}-x_{a,i}\right)}^{2}$$
(6)$$RMSE=\sqrt{\frac{1}{n}\sum_{i=1}^{n}{\left(x_{p,i}-x_{a,i}\right)}^{2}}$$
(7)$$SEP=\frac{RMSE}{x_{a,ave}}\times 100$$
(8)$$MAE=\frac{1}{n}\sum_{i=1}^{n}\left|\left(x_{a,i}-x_{p,i}\right)\right|$$
(9)$$AAD=\frac{1}{n}\left(\sum_{i=1}^{n}\left(\frac{\left|\left(x_{a,i}-x_{p,i}\right)\right|}{x_{a,i}}\right)\right)\times 100$$

In Equations (2)–(9), *n* is the number of experimental runs,$\;x_{p,i}$ is the predicted value, $x_{a,i}$ is the experimental values, $x_{a,ave}$ is the average of the experimental values, $x_{p,ave}$ is the average of the predicted values, and k is the number of input variables.

### 2.4. Optimisation of Responses

An algorithm-based approach was used to determine the optimum levels of the dependent variables and the independent factors. The algorithm functions by searching for a combination of factor levels that simultaneously satisfy the criteria placed on each of the responses and factors. For this work, the built-in optimization algorithms in the Neural Power software were utilized. The optimization algorithms considered were genetic algorithm (GA), particle swarm optimization (PSO), and rotation inherit optimization (RIO).

### 2.5. Preparation of Layered Suppositories

Layered suppositories containing 200 mg of metronidazole and 150 million spores of *Lactobacillus* sp. were prepared by the fusion method. The first layer and third layers contained metronidazole powder incorporated into the melted base. The determined volume of melted mass was poured into the appropriate suppository metal mold and allowed to cool at room temperature. The middle layer was prepared by fusion method, and the determined volume of melted mass containing *Lactobacillus* sp. spores was poured into the middle layer and cooled again to room temperature to produce the second layer with 150 million *Lactobacillus* sp., after which the third layer was introduced. The layered suppositories were weighed and kept at room temperature for 24 h after removal from the mold to allow for uniform solidification, after which they were stored at 4 °C [4]. 

### 2.6. Physicochemical Characterization of the Layered Suppositories

The mechanical strength of the suppositories was determined using a Monsanto hardness tester, and the weight required for the suppository to collapse was recorded in grams. Using the Roches Friabilator^®^, twenty suppositories were evaluated for 4 min at 25 rpm. A loss of less than 1% in weight is generally considered acceptable [17]. Other parameters, such as melting point and disintegration time, were determined in line with USP specifications [4,17]. Twenty suppositories were selected at random to determine the uniformity of weight. The formulation was said to have met the standard if no more than two of the individual weights deviated from the average weight by more than 10% deviation. Content uniformity of 10 suppositories taken at random was carried out, with the concentration of active ingredient spectrophotometrically determined at 320 nm. The formulation was said to be within BP standard if all 10 values fall within 85–115% of the average suppository dose. Assessment for homogeneity was carried out via visual inspection. Ten suppositories were split open longitudinally and assessed visually for bleeding into layers. The batch was seen to have passed if all formulations showed no bleeding, i.e., all the layers were distinctly seen.

### 2.7. Stability Testing of Optimized Suppositories

The optimized layered suppositories were packed in suitable packaging materials and stored under the following conditions for a period as prescribed by International Council for Harmonisation of Technical Requirements for Pharmaceuticals for Human Use (ICH) guidelines for stability studies [18] for a period of 90 days. The suppositories were withdrawn periodically and analyzed for physical characterization (visual defects, mechanical strength, melting, disintegration time and dissolution rate, etc.) and drug content. 

### 2.8. In Vitro Drug Release Analysis of Layered Suppositories

The in vitro release of Metronidazole and *Lactobacillus* spp. from suppositories was performed in triplicate using the U.S.P dissolution apparatus II. The paddle was rotated at 50 rpm in 900 mL of citrate buffer (pH 4.4), maintained at 37 ± 0.5 °C. The amount of drug released was spectrophotometrically determined at 320 nm. 

### 2.9. Lactobacilli Cell Viability of the Formulated Suppositories

The formulated suppositories were incorporated with the *Lactobacillus* load of 10^8^ cells per suppository and, at predetermined time intervals, were evaluated for *Lactobacillus* viability using method of Rodrigue et al. [10]. Briefly, 3% Tween 80 was employed for dilution using a microtiter plate, micropipette, and micro pipettor. Then, 10 µL of a melted suppository containing *Lactobacillus* spp. was added to 90 µL of Tween 80 solution to give appropriate dilution with Tween 80 and was plated using MRS agar. Incubation ensued for 72 h, after which colonies were counted under the colony counter and percentage viability was calculated using Equation (10) where *N*_0_ = total count and *N*_1_ = viable count.
(10)$$\mathrm{Percentage}\;\mathrm{Viability}\;(\%)=\frac{N_{1}}{N_{0}}\times 100$$

### 2.10. Assessment of Vagina Epithelium Exposed to Metronidazole Containing Lactobacillus spp. Formulation

Fourteen female adult healthy treatment naïve rats weighing 160–170 g purchased from Komad^®^ farms in Ikorodu, Lagos State, Nigeria, were utilized in this study. Acclimatization in aerated rooms housing 590 × 400 × 210 mm polycarbonate cages at 29 ± 2 °C, 12 h light/dark cycles, and 40 ± 3%RH was ensured. All the experiments were approved in writing by Health Research Ethical Committee CMUL/HREC/10/19/645. This study utilized the Animal research: reporting in vivo experiments: the ARRIVE guidelines checklist [19] in documenting the study. The rats were randomized into three groups (12 test and 2 control) and were administered daily doses of 3 g/kg-bwt of the formulated suppositories suppository intravaginally. The external appearance of the vagina was observed daily, and visible signs of damage such as difficulty in inoculation, vagina contraction, spasms, redness, or burning were noted. On days 7 and 14, the rats were humanely euthanized (via exposure to carbon dioxide gas), and vaginal tissues were excised and fixed in 10% formalin solution for histological analysis [19].

### 2.11. Statistical Analysis

The data were expressed as mean standard deviation (±SD) using ANOVA (±SD). Significant differences (*p* < 0.05) of mean values were determined using the Tukey test.

## 3. Results

### 3.1. Physicochemical Characterization of the Layered Suppositories

The formulated suppositories had clearly defined layered (yellow and white) with a continuous smooth surface. There was an absence of bleeding from one layer into another. There were no fissures or holes due to suppository contraction. The formulated suppositories met the official British pharmacopeia standard regarding weight and content uniformity. There were no notable differences in color and opacity between prepared suppositories.

### 3.2. ANN Architecture and Training

The best transfer function was chosen as the one with the highest R^2^ value and lowest RMSE value. It is clearly seen from Table 2 that the Gaussian transfer function was the most suitable for the case of drug metronidazole release, while for the case of *Lactobacilli* viability, the hyperbolic tangent transfer function was found to be most suitable. Thus, these two transfer functions were chosen for the networks developed for drug metronidazole release and *Lactobacilli* viability, respectively.

The results of the selection of the best network architecture, as well as the most suitable training algorithm, are presented in Table 3. The best network architecture and corresponding training algorithm were chosen as a multilayer full feed forward neural network and incremental back propagation, and this was chosen because it had the highest R^2^ value and lowest RMSE value for both responses. This network was found suitable for modeling both responses adequately.

Figure 1A,B show the result of the determination of the optimum number of neurons for the network developed for both responses. For the case of drug metronidazole release, Figure 1A shows that as the number of neurons was increased from one to five, the R^2^ value increased, indicating an increase in the predictive capacity of the network [13]. However, increasing the number of neurons beyond five did not improve the R^2^ value, thus suggesting that the optimum number of neurons is five. Following the same reasoning, it can be seen (Figure 1B) that the optimum number of neurons for the case of *Lactobacilli* viability was three. Thus, the configuration of the network for drug metronidazole release was denoted as 2-5-5-1, indicating that the network contains two input variables, five neurons each in both hidden layers, and one output variable. On the other hand, the configuration of the network for *Lactobacilli* viability was denoted as 2-3-3-1, indicating that the network contains two input variables, three neurons each in both hidden layers, and one output variable. These two networks are shown in Figure 1C,D, respectively.

### 3.3. Validation of ANN Model Predictions

The values of drug metronidazole release and *Lactobacilli* viability predicted by the optimum network topologies (Figure 1C,D) are presented in Table 3 alongside the experimental values for comparison. From the data in Table 3, the model predictions were very close to the experimental observations, indicating minimal deviation between them and thus showing the validity of the ANN models. This observation was also corroborated by the goodness of fit statistics presented in Table 4. Generally, the ANN models for drug metronidazole release and *Lactobacilli* viability displayed high predictive capacity as seen in the high R (0.9997 and 0.9999, respectively), R^2^ (0.9995 and 0.9999, respectively) and adjusted R^2^ values (0.9995 and 0.9999, respectively) as well as low MSE (0.0467 and 0.0005, respectively), RMSE (0.2161 and 0.0225, respectively), SEP (0.2427 and 0.0191, respectively), MAE (0.0985 and 0.0052, respectively), and AAD (0.1140 and 0.0169, respectively) values. The higher R, R^2^, and adjusted R^2^ values, as well as the lower MSE, RMSE, SEP, MAE, and AAD values for the ANN model for *Lactobacilli* viability compared with those of the model for drug metronidazole release indicate the superiority of the ANN model for *Lactobacilli* viability in terms of its predictive capability. The parity plots (Figure 2A,B) compare the predicted and actual results for both responses. There was generally an acceptable level of fit between the experimental and model predicted results, as seen in the fact that the data points all clustered around the 45° diagonal line showing that there was minimal deviation between experimental and predicted values, thus indicating optimal validity of the model [12,14].

### 3.4. Effect of Input Factors on the Responses

The 3D response surface plots shown in Figure 2A,B were generated to show the relationship between the input factors (independent variables) and the responses (dependent variables). Figure 2C shows the effect of the Ovucire/PEG ratio and surfactant concentration on drug metronidazole release. Low values of Ovucire/PEG ratio and surfactant concentration maximized the drug metronidazole release. This is because as both Ovucire/PEG ratio and surfactant concentration were increased, the response (drug metronidazole release) was observed to decrease, indicating antagonistic effect of both factors. The *Lactobacilli* viability showed a slightly positive trend with respect to surfactant concentration (Figure 2C,D). The reverse was, however, the case for Ovucire/PEG ratio.

### 3.5. Optimization of Input Factors and Responses

The maximum drug metronidazole release predicted by the three optimization algorithms (GA, PSO, and RIO) was 97.029% for all three cases. The corresponding values of Ovucire/PEG ratio and surfactant concentration for all three cases were 1.086 and 0.046% *w*/*w*, 1.087 and 0.046% *w*/*w*, and 1.087 and 0.046% *w*/*w*, respectively, indicating no difference in the performance of the algorithms (Table 5). The maximum *Lactobacilli* viability predicted by the three optimization algorithms (GA, PSO, and RIO) was 97.410% for all three cases.

The optimized suppositories were developed and evaluated (Figure 3) for metronidazole release and *Lactobacilli* viability, as these were the conditions that were optimized for the layered suppository formulation (Table 6). The formulation had well-defined layers, which consisted of the PEG layer containing metronidazole and the ovucire layer containing the *Lactobacilli* spp. The suppositories had an average weight of 1.17 ± 0.07 g, with a disintegration time of 5.93 ± 0.98 min. In total, 97.63 ± 0.22% of metronidazole was optimally released within 7.5 min, with *Lactobacilli* viability maintained at 97.40%.

### 3.6. Assessment of Vagina Epithelium Exposed to Metronidazole Containing Lactobacillus spp. Formulation

As shown in Figure 4, the vaginal tissue, when treated with the layered suppository showed in Figure 4A normal vagina epithelium (stratified squamous epithelium) (blue arrow), normal vagina stroma (orange arrow), with the normal architecture of the muscularis layer (black arrow) and the adventitia layer (green arrow); Figure 4B shows normal regular vagina epithelium (blue arrow) with the normal architecture of the lamina proper (stroma) (orange arrow). The treatment with the marketed suppository shown in Figure 4C is the normal architecture of the muscularis (black arrow) and the adventitia layer (green arrow). There was an erosion of the stratified epithelium of the vagina (blue arrow) with slight stroma abnormalities (orange arrow). After 14 days, as shown in Figure 4D, massive erosion in the stratified squamous epithelium of the vagina (blue arrow) with normal muscularis layer (black arrow) and lamina propria (stroma) (orange arrow) were seen. Figure 4E,F showed vagina epithelium (stratified squamous epithelium) (blue arrow) and normal vagina stroma (orange arrow), with the normal architecture of the muscularis layer (black arrow) and the adventitia layer (green arrow) in line with the control group which was not challenged with any medication.

### 3.7. Stability Testing

The optimized layered suppositories, which were packed in suitable packaging materials and stored for a period of 90 days, exhibited excellent results for mechanical strength, melting point, disintegration time, and drug content. Storage did not affect any of these parameters with a disintegration time of 6.05 ± 0.18 min; 97.77 ± 0.18% of metronidazole was optimally released within 7.5 min with *Lactobacilli* viability maintained at 97.01% (Table 6).

## 4. Discussion

Bacterial vaginosis remains the commonest vaginal infection among women in their reproductive years [20]. BV is a polymicrobial clinical syndrome characterized by a shift in healthy vaginal microbiota away from *Lactobacillus* species toward more diverse invasive unhealthy bacterial species invasion [21]. Despite the high recommendation of metronidazole as the drug of choice, BV is characterized by high recurrence rates in women with BV due to the allowed persistence of an unfavorable vaginal microbiome, which translates into frequent relapses in addition to the unpleasant side effects [21]. Attempts have been made to reduce the recurrence by use of Probiotics containing suppository with some reported positive outcome. This study focused on the formulation and the enhancement of metronidazole-containing *Lactobacillus* suppository using an optimization technique. The presence of *Lactobacillus* species and other bacterial species in a healthy human vagina contributes to maintaining the 3.5–4.5 acidic pH and produces several antimicrobial substances that inhibit pathogenic organisms. Alterations of normal vaginal microflora are the predisposition to recurrent BV hence the need for formulation of the vaginal medicated suppository that offers replacement therapy of vagina flora.

### 4.1. Physicochemical Characterization of the Layered Suppositories

The formulated vaginal suppositories were characterized with respect to color and opacity, odor, homogeneity, and weight variation. The formulated suppositories were oval shaped, having a white and yellow appearance. All formulated suppositories disintegrated in less than 60 min, with the highest disintegration time at 12.76 ± 0.37 min, as shown in Table 1. The melting point of the suppositories was within compendial limits, with the highest value obtained being 35–37.5 °C; this is in consonance with results obtained by Rodrigues et al. [10] and Sankar et al. [17]. The suppositories with a higher concentration of Ovucire did not show appreciably high solidification points compared with the blend having an equal concentration of PEG and Ovucire, i.e., C and E, as shown in Table 1. Ovucire has an optimized concentration at which the desired solidification points are obtained; increasing the concentration beyond this point created densely formed suppositories that lacked a cohesive structure when the heat was applied hence the absence of an appreciably higher solidification point. The comparative similarity in the density of the PEG blend with Ovucire ensured that the layered suppository possessed consistency and good wetting properties adapted for vaginal drug release and product storage.

### 4.2. Effect of Input Factors on the Responses after Validation Using ANN Model Predictions

There was a good relationship between the experiment and predicted responses with respect to the percentage of metronidazole release and *Lactobacilli* viability. The 3D response surface plots shown in Figure 2A,B were generated to show the relationship between the input factors (independent variables) and the responses (dependent variables). Figure 2C,D show the effect of Ovucire/PEG ratio and surfactant concentration on drug metronidazole release. Low values of Ovucire/PEG ratio and surfactant concentration maximized the drug metronidazole release. This is because as both Ovucire/PEG ratio and surfactant concentration were increased, the wetting properties of the suppository were reduced hence affecting drug release. An increase in surfactant concentration could also lead to dose dumping of the metronidazole, this burst release of loaded metronidazole could also be associated with increased instability of the formulation after disintegration has occurred in the vagina environment. The response (drug metronidazole release) was observed to decrease, indicating antagonistic effect of both factors. Hence, optimization factors were derived to ensure a balance between drug release, which ensures that active pharmaceutical ingredient is optimally released, and *Lactobacilli* viability, which ensure that the environment of the vagina is not unduly eroded whilst treatment of bacterial vaginosis is ongoing. Preservation of vagina microflora is critical to ensure that bacterial clearance is achieved. The *Lactobacilli* viability showed a slightly positive trend with respect to surfactant concentration (Figure 2C). This may be due in part to the amphiphilic surfactant compounds produced by *Lactobacilli* spp., anchored on the surface, or secreted to the outside, with outstanding surface and emulsifying properties. The *Lactobacilli* spp. contain secondary metabolites which engage in the proliferation of host interaction to facilitate biofilm clearance via increasing anti-adhesive tendencies of the bacteria the formulation is designed to tackle. De Gregorio et al. [22] evaluated the antagonist effect of biosurfactants against Candida, and the influence of different concentrations of biosurfactant at 1.25 mg/mL on the adhesion of C. albicans to HeLa cells was investigated using displacement assays. The study showed the potential of biosurfactants in *Lactobacilli* to interfere with and ameliorate Candida adhesion and mucosal inflammation.

### 4.3. Optimization of Input Factors and Responses

The corresponding values of Ovucire/PEG ratio and surfactant concentration for all three cases were 1.042 and 0.000342% *w*/*w*, 1.038 and 0.000113% *w*/*w*, and 1.041 and 0.000483% *w*/*w*, respectively. The performance of the three optimization algorithms was also not significantly different [22,23,24]. In total, 97.63 ± 0.22% of Metronidazole was optimally released within 7.5 min, with *Lactobacilli* viability maintained at 97.40%. The drug release kinetics employed for the metronidazole layered suppositories followed zero order and Higuchi order release model with R2 values of 0.981 and 0.992, respectively. The result of the above findings gave rise to the need for an optimized formulation that will have a maximal release of the drug metronidazole as well as maintain the highest possible *Lactobacilli* viability in the suppositories to combat the BV recurrence. The optimized suppositories were formulated by utilizing a very low surfactant concentration to achieve maximal viability of the *Lactobacilli* in the formulation while still maintaining the low ovucire/PEG ratio and tween 80 for maximum drug metronidazole release. The optimized product formulated using the composite method showed a similar maximal drug metronidazole response irrespective of the experimental model (GA/PSO/ RIO) utilized.

The layered suppository served as a solid matrix from the poorly soluble metronidazole to be released from. A pseudo-steady-state approach utilizing a direct proportionality between the cumulative amount of metronidazole released and the square root of time was observed. The Higuchi and zero-order models effectively described the limits for transport and metronidazole release as being concentration dependent hence zero order kinetics with the constant release over a period, as well as releases from a solid matrix system via the Higuchi order release model.

### 4.4. Assessment of Vagina Epithelium Exposed to Metronidazole Containing Lactobacillus spp. Formulation

Histopathology studies of the vaginal tissues of rats treated with the optimized layered metronidazole suppositories showed no distortion or loss in the vagina epithelium; this was very similar to the control group, hence a strong indicator of the safety of the formulation. The erosion of the stratified epithelium of the vagina seen when conventional metronidazole suppository was utilized can be linked to the absence of *Lactobacilli* spp., hence an absence of the protection *Lactobacilli* spp. offers. *Lactobacilli* are the predominant microorganisms in the vaginal microbiota of healthy women that contribute to preventing urogenital and sexually transmitted infections [18,22]. The protective role of formulations used in treating bacterial vaginosis is critical to ensure that the vagina tissue exhibits competitive exclusion against pathogens, reducing biofilm formation. The marketed formulation preserved the normal architecture of the muscularis of the vagina, but after 14 days, stroma abnormalities were evident.

## 5. Conclusions

To reduce the prevalence of BV and prevent a recurrence, focus on intervention must be holistically taken into consideration to include the use of flora-maintaining metronidazole or antibiotic therapy. This study successfully developed optimized metronidazole containing *Lactobacillus* suppository using artificial neural networks, which is intended to treat BV by eradicating bacterial pathogens as well as restoration of the normal vaginal microbiota. The formulation was found to be non-toxic and safe for the vaginal epithelium of test animals. The optimized formulation seems to be very promising, allowing to achieve a nonsignificant loss of viable bacteria necessary for the treatment of recurrent bacterial vaginosis.

## Figures and Tables

**Figure 1 pharmaceutics-14-02337-f001:**
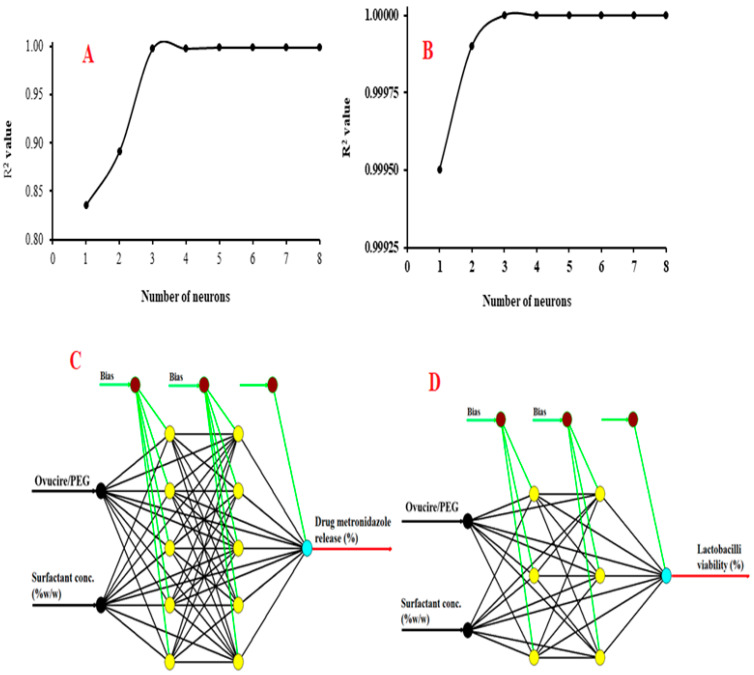
(**A**) Selection of optimum number of neurons for drug metronidazole release (**B**) Selection of optimum number of neurons for *Lactobacilli* viability (**C**) Architecture of the optimal ANN for predicting drug metronidazole release (**D**) Architecture of the optimal ANN for predicting *Lactobacilli* viability.

**Figure 2 pharmaceutics-14-02337-f002:**
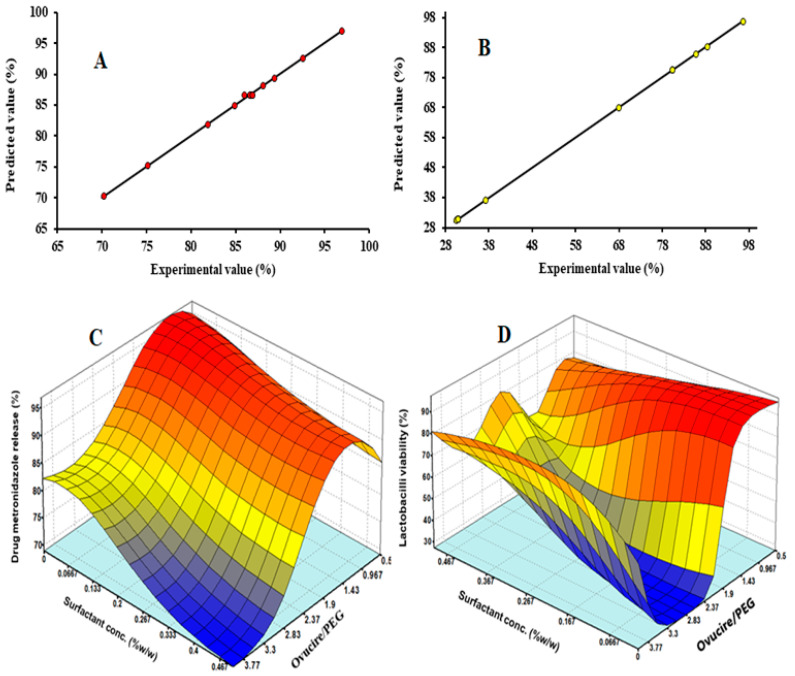
(**A**) Parity plot showing comparison of experimental and predicted drug metronidazole release (**B**) Parity plot showing comparison of experimental and predicted *Lactobacilli* viability (**C**) Response surface plot showing effect of Ovucire/PEG ratio and surfactant concentration on drug metronidazole release (**D**) Response surface plot showing effect of Ovucire/PEG ratio and surfactant concentration on *Lactobacilli* viability.

**Figure 3 pharmaceutics-14-02337-f003:**
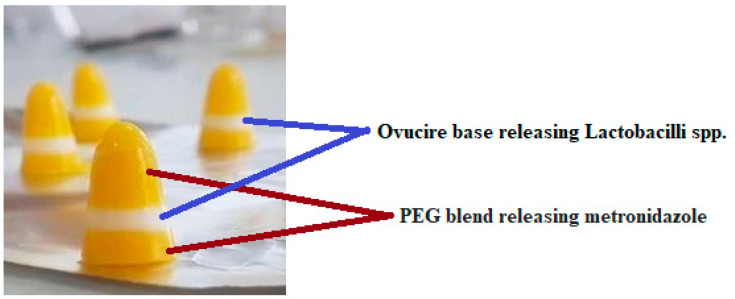
Multi-layered suppository containing metronidazole and *Lactobacillus* sp.

**Figure 4 pharmaceutics-14-02337-f004:**
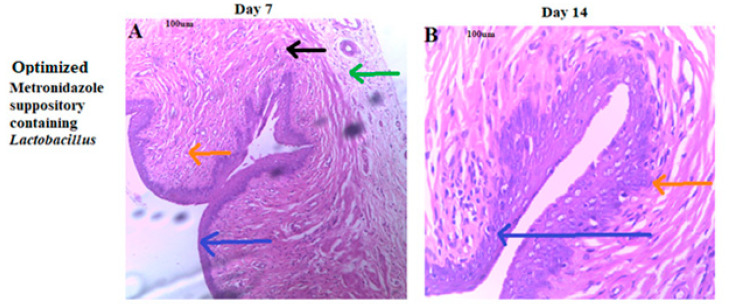
Photomicrograph after days 7 and 14 of histology slides of the vaginal tissue treated with varying (**A**,**B**) optimized metronidazole suppository containing *Lactobacilli*, (**C**,**D**) marketed metronidazole suppository, (**E**,**F**) control without any treatment.

**Table 1 pharmaceutics-14-02337-t001:** Physical characteristics of Metronidazole and *Lactobacillus* layered suppositories.

Physical Characteristics	A	B	C	D	E	F	G	H	I	J	K	L	M
Color and Opacity	Yellow and White Opaque												
Shape	Conical												
Homogeneity	Homogenous												
Weight Variation * (gram)	1.17 ± 0.05	1.14 ± 0.05	1.2 ± 0	1.19 ± 0.03	1.13 ± 0.05	1.13 ± 0.05	1.12 ± 0.04	1.12 ± 0.04	1.16 ± 0.05	1.19 ± 0.03	1.19 ± 0.03	1.13 ± 0.09	1.14 ± 0.05
Hardness * (kilogram)	0.8 ± 0.28	0.9 ± 0.42	1.3 ± 0.14	1.0 ± 0.0	1.0 ± 0.0	0.8 ± 0.28	0.7 ± 0.14	0.6 ± 0	0.9 ± 0.14	0.8 ± 0	0.74 ± 0.14	0.8 ± 0.28	0.6 ± 0.0
Melting Point range (°C)	32–36.5	33.5–38	35–38	31.5–37	32.5–377	32–36.5	32–36.5	30.45–37	32–36.5	32–37	35–37.5	34–36.5	32–36.5
Solidification Point * (°C)	38.5 ± 0.71	34.5 ± 0.71	38.9 ± 1.56	39.5 ± 4.95	37.4 ± 0.91	36.0 ± 1.41	36.5 ± 2.12	35.0 ± 1.41	36.5 ± 0.71	37.5 ± 2.12	39.0 ± 1.41	39.0 ± 1.41	36.5 ± 0.71
Disintegration time * (minutes)	10.83 ± 0.56	10.84 ± 0.45	12.76 ± 0.37	10.78 ± 0.69	10.55 ± 0.04	11.02 ± 0.01	10.07 ± 0.09	7.55 ± 0.02	9.95 ± 0.35	10.65 ± 0.70	11.29 ± 0.23	10.2 ± 0.13	9.87 ± 0.99

* Result represented as (*n* = 6; Mean ± SD).

**Table 2 pharmaceutics-14-02337-t002:** Selection of best transfer function.

Transfer Function	Drug Metronidazole Release (%)		*Lactobacilli* Viability (%)	
	R^2^	RMSE	R^2^	RMSE
Sigmoid	0.9767	1.0436	0.9998	0.3485
Tanh	0.9989	0.2297	**0.9999**	**0.0225**
Gaussian	**0.9990**	**0.2161**	0.9992	0.2548
Linear	0.8304	2.8136	0.1631	25.1260
Threshold linear	0.0014	6.8313	0.6115	17.1200
Bipolar linear	0.0014	6.8313	0.6642	15.9150

**Table 3 pharmaceutics-14-02337-t003:** Selection of best network architecture.

Network Architecture	Training Algorithm	Response			
		Drug Metronidazole Release (%)		*Lactobacilli* Viability (%)	
		R^2^	RMSE	R^2^	RMSE
MNFF	IBP	0.9990	0.2161	0.9999	0.0225
	BBP	0.8644	2.5152	0.9999	0.0250
	QP	0.9889	0.7208	0.9999	0.1919
	GA	0.9861	0.8067	0.9999	0.0337
	LM	0.3402	5.5489	0.0434	26.8630
MFFF	IBP	**0.9995**	**0.2161**	**1.0000**	**0.0225**
	BBP	0.9972	0.3590	1.0000	0.0232
	QP	0.9661	1.2581	1.0000	0.0233
	GA	0.9990	0.2162	0.9999	0.0337
	LM	0.9990	0.2162	0.0434	26.8630

**Table 4 pharmaceutics-14-02337-t004:** Comparison of experimental results with ANN predicted results.

Run	Blends	Factors		Responses			
				Drug Metronidazole Release (%)		*Lactobacilli* Viability (%)	
		Ovucire/PEG	Surfactant Concentration (%*w*/*w*)	Experiment	Predicted	Experiment	Predicted
1	A	2.25	0.25	86.64	86.59	30.96	30.95
2	B	3.49	0.07	81.85	81.85	37.21	37.21
3	C	1.01	0.07	96.91	96.91	96.64	96.64
4	D	2.25	0.25	85.95	86.59	30.96	30.95
5	E	1.01	0.43	92.54	92.54	80.45	80.45
6	F	2.25	0.25	86.64	86.59	30.96	30.95
7	G	4.00	0.25	75.17	75.17	88.34	88.34
8	H	3.49	0.43	70.28	70.28	67.92	67.92
9	I	2.25	0.50	84.90	84.90	80.44	80.44
10	J	0.50	0.25	89.37	89.37	85.71	85.71
11	K	2.25	0.25	86.86	86.59	30.98	30.95
12	L	2.25	0.00	88.10	88.10	30.57	30.57
13	M	2.25	0.25	86.86	86.59	30.88	30.88

**Table 5 pharmaceutics-14-02337-t005:** Goodness of fit statistics for ANN model.

Parameter	Response	
	Drug Metronidazole Release	*Lactobacilli* Viability
*R*	0.9997	0.9999
*R* ^2^	0.9995	0.9999
Adjusted *R*^2^	0.9995	0.9999
*MSE*	0.0467	0.0005
*RMSE*	0.2161	0.0225
*SEP*	0.2427	0.0191
*MAE*	0.0985	0.0052
*AAD*	0.1140	0.0169

**Table 6 pharmaceutics-14-02337-t006:** Optimized conditions for drug metronidazole release and *Lactobacilli* viability.

Variables	Optimization Algorithms					
	GA		RIO		PSO	
	Drug Metronidazole Release (%)	*Lactobacilli* Viability (%)	Drug Metronidazole Release (%)	*Lactobacilli* Viability (%)	Drug Metronidazole Release (%)	*Lactobacilli* Viability (%)
Ovucire/PEG	1.086	1.042	1.087	1.038	1.087	1.041
Surfactant concentration (%*w*/*w*)	0.046	0.000342	0.046	0.000113	0.046	0.000483
Predicted maximum response value (%)	97.029	97.410	97.029	97.410	97.029	97.410
Actual maximum response value (%)	97.63 ± 0.22	97.40				

## Data Availability

Not applicable.

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
