# Peer review of "Formulation and Optimization of Metronidazole and Lactobacillus spp. Layered Suppositories via a Three-Variable, Five-Level Central Composite Design for the Management of Bacterial Vaginosis"

_pharmaceutics, 2022, doi:10.3390/pharmaceutics14112337_

Round 1

Reviewer 1 Report

the manuscript by Ilomuanya et.al. describing optimization of layered suppository containing metronidazole and lactobascillus is extremely interesting and relevant given the grave health conditions of women due to VB infections and subsequent complications arising from existing antibiotic treatments. 
This author believes that this manuscript may be published in the present form.  

Author Response

Reviewer 1

  • Reviewers Comment: the manuscript by Ilomuanya et.al. describing optimization of layered suppository containing metronidazole and lactobascillus is extremely interesting and relevant given the grave health conditions of women due to VB infections and subsequent complications arising from existing antibiotic treatments. 

Authors Response: Manuscript has been corrected for English language where applicable.

  • Reviewers Comment: This reveiwer believes that this manuscript may be published in the present form

Authors Response:.  Author agrees with the reviewer

Reviewer 2 Report

Formulation and Optimization of Metronidazole and Lactobacillus Spp Layered Suppositories via a three-variable, five-level 3 central composite design for the management of bacterial vaginosis

The manuscript has been written and presented well however, the following are some points that need to be addressed by the authors before it could be considered for publication.

Line 15

Abstract: The availability of a vaginal layered suppository with an antimicrobial agent and a probiotic has the potential to provide a synergistic effect and prevent recurrence in the management of bacterial vaginosis.

Please rephrase the title and this sentence to reflect the studies undertaken in this manuscript.

Line 27

This optimised formulation provides a safe, and promising alternative to conventional suppositories in the treatment and prevention of recurrence of bacterial vaginosis as it synergistically tackles the infectious microbe while restoring and maintaining the normal vaginal flora.

Please rephrase this sentence to reflect the studies undertaken in this manuscript.

Did authors prove synergistic effect?

`Line 162

mass containing Lactobacillus sp spores was poured into the middle layer and cooled again to room temperature to produce the second layer with 150 million Lactobacillus sp

Please add details of the third layer.

Line 176

Melting point is a measure of the time taken for the entire suppository to melt or disperse when immersed in a water bath maintained at 37 ± 1oC.

Please add relevant citation

Line 192

performed using the U.S.P 25 dissolution apparatus II. The paddle was rotated at 50 rpm in 900 mL of citrate buffer (pH 4.4), maintained at 37 ± 0.5°C.

Please add relevant citation.

Please describe why citrate buffer was used.

Line 220

daily doses of 3g/kg-bwt of the formulated suppositories suppository and a marketed brand intravaginally

Please correct.

Line 225

were humanely euthanized (via exposure to carbon dioxide gas) and vaginal tissues were excised and fixed in 10% formalin solution for histological analysis.

Please add relevant citation

Line 232

3.1. Physicochemical Characterization of the bilayer suppositories

Its bilayer or multilayer?

Line 235

The formulated suppositories met official British pharmacopoeia standard regarding weight and content uniformity.

Please add methodology details in method section with relevant citations.

Line 363

stored for a period of 90 days exhibited excellent results for mechanical strength, melting point, disintegration time and drug content.

Please cite table or figures of these results.

Line 366

optimally released within 7.5 minutes with lactobacilli viability maintained at 97.01%.

Please cite table or figures of these results.

Line 384

Physicochemical characterization of the bilayer suppositories

Bilayer or multilayer

Line 386

and opacity, odor, homogeneity, and weight variation.

Please describe how homogeneity was measured in methods.

Line 390

e highest value obtained being 35 – 37.5 oC 20

Please check and correct

Line 392

higher concentration of Ovucire did not show appreciably high solidification points

Please suggest why higher concentration of Ovucire did not show appreciably high solidification points

Author Response

Formulation and Optimization of Metronidazole and Lactobacillus Spp Layered Suppositories via a three-variable, five-level 3 central composite design for the management of bacterial vaginosis

The manuscript has been written and presented well however, the following are some points that need to be addressed by the authors before it could be considered for publication.

AUTHORS RESPONSE TO REVIEWER’S COMMENTS

Line 15

Abstract: The availability of a vaginal layered suppository with an antimicrobial agent and a probiotic has the potential to provide a synergistic effect and prevent recurrence in the management of bacterial vaginosis.

Please rephrase the title and this sentence to reflect the studies undertaken in this manuscript.

This section now reads “Bacterial vaginosis, a polymicrobial clinical syndrome characterized by a shift in healthy vaginal microbiota due to bacterial colonization is characterized by high recurrence rates after conventional treatment an antimicrobial agent.  This has necessitated the need to develop a formulation that has the potential to ensure lactobacilli viability and bacterial clearance.  

Line 27

This optimised formulation provides a safe, and promising alternative to conventional suppositories in the treatment and prevention of recurrence of bacterial vaginosis as it synergistically tackles the infectious microbe while restoring and maintaining the normal vaginal flora.

Please rephrase this sentence to reflect the studies undertaken in this manuscript.

Did authors prove synergistic effect?

This has been edited to read “This optimised formulation provides a safe, and promising alternative to conventional suppositories in the treatment and prevention of recurrence of bacterial vaginosis”

`Line 162

mass containing Lactobacillus sp spores was poured into the middle layer and cooled again to room temperature to produce the second layer with 150 million Lactobacillus sp

Please add details of the third layer.

This has been included

Line 176

Melting point is a measure of the time taken for the entire suppository to melt or disperse when immersed in a water bath maintained at 37 ± 1oC.

Please add relevant citation

The citation for this section has been added and the entire section re-written

Line 192

performed using the U.S.P 25 dissolution apparatus II. The paddle was rotated at 50 rpm in 900 mL of citrate buffer (pH 4.4), maintained at 37 ± 0.5°C.

Please add relevant citation.

Please describe why citrate buffer was used.

The citation for this section has been added. Citrate buffer was used to ensure a simulation with the vaginal fluid

Line 220

daily doses of 3g/kg-bwt of the formulated suppositories suppository and a marketed brand intravaginally

Please correct.

This has been corrected to read “The rats were randomized into three groups (12 test and 2 control) and were administered daily doses of 3g/kg-bwt of the formulated suppositories suppository intravaginally.”

Line 225

were humanely euthanized (via exposure to carbon dioxide gas) and vaginal tissues were excised and fixed in 10% formalin solution for histological analysis.

Please add relevant citation

This has been included

Line 232

3.1. Physicochemical Characterization of the bilayer suppositories

Its bilayer or multilayer?

This has been corrected through the entire manuscript to read layered

Line 235

The formulated suppositories met official British pharmacopoeia standard regarding weight and content uniformity.

Please add methodology details in method section with relevant citations.

This has been included in the manuscript

Line 363

stored for a period of 90 days exhibited excellent results for mechanical strength, melting point, disintegration time and drug content.

Please cite table or figures of these results.

This is reflected in the results section 3.7

“The optimized layered suppositories which were packed in suitable packaging materials and stored for a period of 90 days exhibited excellent results for mechanical strength, melting point, disintegration time and drug content. Storage did not affect any of these parameters with disintegration time of 6.05 ± 0.18 minutes; 97.77% ± 0.18% of metronidazole was optimally released within 7.5 minutes with lactobacilli viability maintained at 97.01%.”

Line 366

optimally released within 7.5 minutes with lactobacilli viability maintained at 97.01%.

Please cite table or figures of these results.

This has been included

Line 384

Physicochemical characterization of the bilayer suppositories

Bilayer or multilayer

Layered

Line 386

and opacity, odor, homogeneity, and weight variation.

Please describe how homogeneity was measured in methods.

This has been added into the manuscript

Line 390

e highest value obtained being 35 – 37.5 oC 20

Please check and correct

This has been corrected

Line 392

higher concentration of Ovucire did not show appreciably high solidification points

Please suggest why higher concentration of Ovucire did not show appreciably high solidification points

This has been included in the manuscript to read “Ovucire has an optimized concentration at which the desired solidification points is obtained increasing the concentration beyond this point created densely formed suppositories which lacked a cohesive structure when heat is applied hence the absence of an appreciably higher solidification point.”

Reviewer 3 Report

I read your excellent R&D papers. I want to congratulate you on an exciting development. There are some unexplained parts, so I have a question.

1. Why was Lactobacillus acidophilus  selected as the sole strain for the multiple lactobacillus strain used together with Metronidasole?

2. Why didn't the previously studied strains known to be effective against bacterial vaginitis(BV), such as Lactobacillus Rhamnosus or L. Reuteri, be used together?

REF) Randomized Controlled Trial Microbes Infect . 2006 May;8(6):1450-4.

3. Is there any evidence to prove which method is more effective: the method of using Metronidazole alone and then using the lactobacillus vaginal suppository or the method of mixing the two?

4.When administered together with metronidazole and lactobacillus strains may also be affected by metronidazole, but the discussion on this issue and the explanation of dose selection is insufficient. 

REF) Infect Dis Obstet Gynecol. 2001; 9(1): 41–45.

Author Response

Reviewers Comments:I read your excellent R&D papers. I want to congratulate you on an exciting development. There are some unexplained parts, so I have a question.

  1. Why was Lactobacillus acidophilus selected as the sole strain for the multiple lactobacillus strain used together with Metronidazole?

Authors response: This is an error and this has been corrected to read Lactobacilli fermentum.

This strain was cultured an typed in my laboratory. The DNA samples showed amplification for the band sharing frequency (BSF) primers producing an amplification size of 526bp by all lactic acid bacteria, this has been published Mucoadhesive Microspheres of Maraviroc and Tenofovir Designed for Pre-Exposure Prophylaxis of HIV-1: An in vitro Assessment of the Effect on Vaginal Lactic Acid Bacteria Microflora - PMC (nih.gov)

  1. Why didn't the previously studied strains known to be effective against bacterial vaginitis(BV), such as Lactobacillus Rhamnosus or L. Reuteri, be used together?

Authors response: We have previously cultured Lactobacillus fermentum and have optimized this bacteria for formulation within a suppository.

REF) Randomized Controlled Trial Microbes Infect . 2006 May;8(6):1450-4.

  1. Is there any evidence to prove which method is more effective: the method of using Metronidazole alone and then using the lactobacillus vaginal suppository or the method of mixing the two?

Authors response: Utilization of metronidazole alone accounts for over 50% recurrence rate with conventional Metronidazole treatment alone (Centre for Disease Control and Prevention (CDC 2020). Bacterial Vaginosis [BV] Statistics http://www.cdc.gov/std/bv/stats.html). Metronidazole and probiotics taken separate account for better prognosis. How ever there is no information of optimization using artificial neural networks of formulation characteristics to ensure viability of Lactobacillus spp within a suppository. This is what this research has carried out.

4.When administered together with metronidazole and lactobacillus strains may also be affected by metronidazole, but the discussion on this issue and the explanation of dose selection is insufficient. REF) Infect Dis Obstet Gynecol. 2001; 9(1): 41–45.

Authors response: Dose selection of metronidazole is optimized and has followed current chemotherapeutic dosing for treatment of BV (the suppository molds used in formulation were 1g molds). The novelty of the formulation is to be able to ensure viability of Lactobacillus spp., ensure its release as well as its activity within the novel layered suppository. This has been achieved used artificial neural networks. The lactobacillus strain is not affected by metronidazole due to the preparation methods and its inherent protection by the polymer in the middle layer of the suppository.